# P188 Therapy in In Vitro Models of Traumatic Brain Injury

**DOI:** 10.3390/ijms24043334

**Published:** 2023-02-07

**Authors:** Michael Zargari, Luise J. Meyer, Matthias L. Riess, Zhu Li, Matthew B. Barajas

**Affiliations:** 1Vanderbilt University School of Medicine, Nashville, TN 37232, USA; 2Albertinen Krankenhaus, 22457 Hamburg, Germany; 3TVHS VA Medical Center, Anesthesiology, Nashville, TN 37212, USA; 4Department of Anesthesiology, Vanderbilt University Medical Center, Nashville, TN 37232, USA; 5Department of Pharmacology, Vanderbilt University, Nashville, TN 37232, USA

**Keywords:** cell membrane stabilizer, concussion, copolymer, neuroprotection, Poloxamer 188, TBI

## Abstract

Traumatic brain injury (TBI) is a significant cause of morbidity and mortality worldwide. Varied mechanisms of injury contribute to the heterogeneity of this patient population as demonstrated by the multiple published grading scales and diverse required criteria leading to diagnoses from mild to severe. TBI pathophysiology is classically separated into a primary injury that is characterized by local tissue destruction as a result of the initial blow, followed by a secondary phase of injury constituted by a score of incompletely understood cellular processes including reperfusion injury, disruption to the blood-brain barrier, excitotoxicity, and metabolic dysregulation. There are currently no effective pharmacological treatments in the wide-spread use for TBI, in large part due to challenges associated with the development of clinically representative in vitro and in vivo models. Poloxamer 188 (P188), a Food and Drug Administration-approved amphiphilic triblock copolymer embeds itself into the plasma membrane of damaged cells. P188 has been shown to have neuroprotective properties on various cell types. The objective of this review is to provide a summary of the current literature on in vitro models of TBI treated with P188.

## 1. Introduction

Traumatic brain injury (TBI) is a significant cause of morbidity and mortality, accounting for approximately 220,000 hospitalizations and 64,000 deaths in the United States annually [1]. TBI most frequently occurs from falls; however, attempted suicide, motor vehicle collisions, and assault are also common causes [1]. Despite it being a ‘silent killer’ at times, this disease has garnered increasing attention in the media and non-medical outlets with a focus on sports-related injuries, particularly from American football and combat sports [2]. The TBI patient population will have a variable clinical presentation depending upon the force and nature of their injury, closed versus penetrating. The degree of injury may be categorized as mild, moderate, or severe and there are heterogenous grading criteria from several organizations including but not limited to the Centers for Disease Control definition published in 2003, the World Health Organization Collaborating Centre Task Force on Mild Traumatic Brain Injury guidelines published in 2005, and the Department of Veterans Affairs criteria updated in 2016. Assessment criteria often use the Glasgow Coma Scale which assesses eye opening, verbal, and motor responses as well as evaluation for focal neurologic deficits, loss of consciousness, presence of amnesia, and subjective symptoms such as feeling dazed, confused, disoriented, or slowed thinking [3]. Other diagnostic modalities include serum biomarkers such as glial fibrillary acidic protein, S100 calcium-binding protein beta and ubiquitin carboxy-terminal hydrolase L1, and neuroimaging with evidence of structural brain injury on computed tomography or magnetic resonance imaging [4]. 

TBI-induced deficits may be dynamic as a primary injury initiates a cascade of secondary damage. Initial insult stems from local tissue destruction with associated axonal transection, neuronal and glial damage, altered perfusion, and even intracerebral hemorrhage leading to parenchymal dysfunction. However, the secondary phase which takes place in the coming days to weeks is constituted by a multitude of incompletely understood cellular processes such as reperfusion injury, disruption to the blood-brain barrier, metabolic dysregulation, and mitochondrial dysfunction including oxidative stress and calcium dysregulation [5]. If severe enough, resultant neuronal death can worsen the outcome beyond the initial injury and result in chronic neurological deficits from impaired synaptic circuits, transmission, and plasticity [6]. Mitigation of primary damage is limited to preventative measures. Thus, recent research efforts have been focused on improving treatment outcomes by developing a more complete understanding of the various cellular processes that comprise the secondary phase and identifying novel therapeutic interventions [6]. Targets of this inquiry include prevention of edema, inflammation, excitotoxicity, glial proliferation, and oxidative stress. Edema alters perfusion by increasing intracranial pressure and is associated with blood-brain barrier disruption as well as cell swelling and death [7]. 

The objective of this review is to provide a comprehensive summary of the current literature on in vitro models of TBI treated with P188. While not novel as an ischemia reperfusion therapeutic, P188’s application to this pathophysiology has made great strides in recent years. An overview of currently used in vitro models, and outcome variables will supply a basis for this review. After briefly touching on several experimental treatments being evaluated, a focus onto P188, its history, mechanism, and role in similar pathophysiology to TBI will occur. A breakdown of literature of P188 in TBI models by relevant cell type will highlight P188 effectiveness across cell lines that are present in the neurovascular unit of the blood-brain barrier, a summary of which can be seen in Table 1. Potential future directions will highlight expansion of current science to similar molecules and improved models. 

### 1.1. In Vitro Models of Traumatic Brain Injury

Despite decades of research on TBI, there is still an incomplete understanding of the various molecular mechanisms that underlie its complex pathophysiology, hindering the development of novel medical therapies. While in vivo models may be more suitable to recapitulate the complexity of TBI, in vitro models offer a unique advantage when attempting to understand and quantify the functional and structural pathology on a cellular level. Additionally, in vitro modeling of TBI allows for a more controlled study environment in contrast to in vivo experimentation which has many potential confounding variables, such as oxygen saturation or hyperoxia, anesthetics, analgesics, and hemodynamic variation [28]. Moreover, in vitro studies are inexpensive and considered more ethically permissible [29]. The development of clinically representative in vitro TBI models continues to be a significant obstacle in the identification of novel compounds. Thus, we will provide a brief overview of the main in vitro models along with commonly measured outcome variables associated with TBI.

Models that are currently being used include axon transection to model penetrating injuries, fluid shear stress to study the altered cellular morphology seen in TBI, shear strain to model non-penetrating TBI, stretch injury to model in vivo axonal deformation, and compression to model the area of focal injury. Hydrostatic pressure models have been developed to produce pressure waves via weight drop to study closed head TBI.

Blunt-force TBI may lead to acceleration/deceleration-induced axonal injury via tension resulting in stretch, shear force, and compression but rarely complete axonal transection [30]. Stretch injury mechanisms are the most used insults in models mimicking blunt force TBI. In these models, cells are attached to a flexible and deformable surface, and this process has even been adapted for high-throughput using 96-well plates. Mechanical strain injury can be fine-tuned using computer controller systems, such as adjustment of vacuum-induced strain on a cell monolayer. Variable insults in strain models include percent, rate and duration of strain, static vs. dynamic strain, and uniaxial vs. biaxial strain [30].

Compression-based tools commonly utilize flat weights dropped from a set height onto dissociated or confluent cells. Height, weight, and duration of the weight compressing cells are titratable variables depending on the degree of desired injury. Microfluidic devices are a more recent development which utilize a nickel-iron micro hammer to compress individual cells with 10%–90% strain, across a wide range of strain rates with high throughput up to 12,000 cells per minute [31].

Transection models utilize either ex vivo tissue or isolated axonal tracts and perform transections utilizing varied tools from laser to scalpel to plastic stylet. Scratch injury models aim to induce a secondary injury that is less severe than transection [30].

Blast-induced traumatic brain injury (bTBI) is the result of exposure to an explosive event leading to pressure changes in the skull. Many of the energy transfer modes are similar to blunt-force trauma, with compression of skull plates, rapid sudden head movement leading to shearing axonal injury, and pressurization and displacement of intravascular blood leading to IR injury [32]. However, it is hypothesized that bTBI also causes microbubbles to form and that these microbubbles collapse with high enough pressure to cause local damage to brain tissue or disruption of membrane integrity such as in the blood-brain barrier. Blast-induced TBI has been successfully modeled by exposing cell cultures to transient microbubbles which collapse, leading to microcavitation [28]. Kanagaraj et al. and Sun et al. utilized an electrical discharge system to create microbubble generating shockwaves [14].

Many of these models can be combined with IR insult to mimic the multifactorial injury in clinical TBI. Most in vitro studies utilize immortalized cell lines such as PC2/PC12 and dissociated primary cultures, however, these approaches lack fidelity in recapitulating the neural micro-environment [29]. Advances in 3D cultures have enabled researchers to replicate tissue stiffness, model the extracellular matrix, and even study cell–cell crosstalk in non-dissociated models [33].

Commonly measured outcome variables in these models employ the use of cell and mitochondrial viability assays. Viable cells can be quantified at the end of an experiment via fluorescence with reagents such as PrestoBlue or LIVE/DEAD viability kits (ThermoFisher Scientific Inc., Waltham, MA, USA). Mitochondrial viability can be assessed in a similar manner with other commercially available assays. Lactate dehydrogenase (LDH) is released when the plasma membrane is damaged and leaks from the cytoplasm into culture medium. This can then be quantified using the formation of formazan, absorbance 490 nm. Cell membrane injury can be further assessed by the uptake of regularly impermeant fluorescent styryl dye FM1-43 into the damaged membrane. Caspase 3 activation, cytochrome c release, and p38 mitogen-activated protein kinase (MAPK) are all used to determine the activity of various apoptotic signaling pathways. Matrix metalloproteinase (MMP) activity has been linked to blood-brain barrier damage and while it is hypothesized that MMPs play a direct role in the permeabilization of the blood-brain barrier, this has not been definitively shown in vivo [34]. Finally, LC3-II and Beclin-1 are measures of autophagy activation and calcium influx may be indicative of neuronal dysregulation [35,36].

### 1.2. Current Treatments for Traumatic Brain Injury

There are currently no effective pharmacological treatments in wide-spread use for TBI, in large part due to challenges that are associated with the development of clinically representative in vitro and in vivo models [37,38]. While demonstrating preclinical efficacy, erythropoietin failed to translate to human studies [39]. A systematic review concluded that statins mitigate adverse cognitive outcomes that are associated with multiple dementia subtypes secondary to TBI, however larger studies are required in order to establish clinical efficacy before adopting its widespread usage [40]. Progesterone, a former front-runner in TBI therapy, demonstrated positive effects in two Phase II clinical trials; however, multiple Phase III clinical trials showed no benefit of progesterone administration in TBI [41]. Studies of other therapies such as NAC and etanercept require further validation with preclinical and clinical trials in order to establish a mechanism of action and effect on TBI [42,43].

Both fenofibrate and minocycline have been shown to decrease swelling in TBI [44,45]. Pioglitazone, rosiglitazone, and N-acetyl cystine (NAC) have demonstrated positive effects via anti-inflammatory and antioxidant actions in TBI preclinical models. Modulation of glial proliferation with cyclic-dependent kinase inhibition using flavopiridol or roscovitine has demonstrated promising preclinical efficacy [6].

### 1.3. Therapeutic Potential of Poloxamer 188 as a Membrane Resealant

In recent years, much attention has been paid to Poloxamer 188 (P188), a Food and Drug Administration (FDA)-approved amphiphilic triblock copolymer of 8.4 kDa, comprised of a hydrophobic center made of 30 polypropylene oxide (PPO) units that are flanked on both sides by 75 hydrophilic polyethylene oxide (PEO) units allowing it to embed itself into the plasma membrane of damaged cells [46,47]. P188 does not undergo metabolism in vivo and is renally eliminated unchanged [48].

P188 has been trialed as an intervention for pathologies across several tissue types and it has been studied in vitro, in vivo, as well as in clinical trials [49]. P188 was first that was approved by the FDA for its use in reducing viscosity of blood before transfusions over 60 years ago and later underwent trials evaluating its use in vaso-occlusive crises associated with sickle cell disease [8]. Other clinical trials investigating P188’s effects include investigation of striated muscle injury protection in Duchenne Muscular Dystrophy patients evaluating respiratory, cardiac, and skeletal muscle outcomes as P188 has been shown to decrease calcium overload and tension in dystrophic cardiomyocytes [9,50]. P188 has been given in healthy volunteers as well as elderly and sick patients in extremis with exceptional safety. A small signal of acute kidney injury in those with pre-existing renal dysfunction had been suggested but was determined to be due to impurities and future purified formulations demonstrated recovery of renal safety [50]. Preclinical studies have demonstrated P188’s cardioprotective effect in various IR injury models [10,11,51]. The utility of copolymer-based cell membrane stabilizers in IR injury occurs through structural cell protection. IR injury leads to damages in membranes and permeable pores, and copolymer-based cell membrane stabilizers act as plugs in these holes achieving membrane integrity, preventing pathologic ion flux and allowing for the activation of endogenous repair mechanisms, Figure 1, [47]. In one study of porcine ST segment myocardial infarction, intracoronary P188 given upon reperfusion improved mitochondrial viability and oxidation and reduced infarct size and troponin leak [13]. However, not all studies have demonstrated a protective effect in cardiac IR injury. No rescue of mitochondrial function was found in an ex vivo cardiac IR model post that was conditioned with P188 [12].

P188 has been shown to have neuroprotective properties on various cell types including endothelial cells, neurons, and glial cells in stroke, Parkinson’s Disease, Amyotrophic Lateral Sclerosis, and mechanically-induced TBI models [46]. The safety of P188 and its efficacy as a membrane resealant has been investigated in cardiovascular and central nervous system diseases, making it an attractive therapeutic target in the treatment of TBI [46,49,51,52,53].

## 2. Astrocyte, Endothelial Cell, and Neuronal In Vitro Models of Traumatic Brain Injury Treated with P188

### 2.1. Astrocytes

The three main components of the neurovascular unit for mimicking blood-brain barrier parenchyma in vitro include microvasculature (endothelial cells), astrocytes, and neurons [54]. Astrocytes act as the buffer between the vascular structure and neurons in the blood-brain barrier. They may have end-feet abutting vasculature and modulate transport across the blood-brain barrier and are particularly important in volume and ion regulation [54]. Astrocytes also upregulate blood-brain barrier functions such as tightness of tight junction, polarity of transporters and specialized metabolic barriers [54]. Astrocytes are affected by inflammation and bradykinin which can activate nuclear factor κB which upregulates interleukin-6 which may act upon endothelium in a positive feedback loop [54]. In fact, the degree of reactive astrogliosis reflects the severity of injury following TBI [55].

Microcavitation from bubbles in a blast TBI model demonstrated astrocyte damage evidenced by decreased cell viability, and increased membrane permeability, calcium dysregulation, and superoxide levels. Astrocytes that were exposed post-injury to P188 for either 3 or 24 h showed improved cell viability, calcium handling, and ROS production, although P188 exposure period did not significantly affect results [14].

In a follow-up study, Chen et al. were able to determine that in mouse C8-D1A astrocytes, microcavitation abolishes normal calcium spiking activity through calcium influx via N-type calcium channels, leading to caspase 3 activation [15]. Restoration of calcium spiking was demonstrated in astrocytes that were incubated with P188 or an N-type calcium channel blocker, suggesting that P188’s neuroprotective properties may be partially mediated through preservation of calcium spiking by resealing damaged N-type calcium channels. Further research is indicated in order to elucidate the relationship between calcium spiking activity and caspase 3 mediated apoptotic signaling. These findings are consistent with other studies demonstrating P188 inhibition of calcium influx in adult mouse cardiomyocytes suffering reoxygenation injury [10].

### 2.2. Endothelial Cells

Endothelial cells are integral to the function of the blood-brain barrier, and their detachment and disruption alters the function of the tight junctions in the blood-brain barrier which increases the permeability of injurious substances. Lotze et al. investigated the effect of P188 on mouse brain microvascular endothelial cells in a combined model of simulated IR and compression (TBI) [16,56]. Mild injury models underwent 1 h of compression at the start of a 5-h hypoxia/2-h reoxygenation exposure. Cells were treated with P188 during the 2-h reoxygenation and demonstrated increased cell viability, metabolic activity, and nitric oxide production as well as decreased membrane damage compared to polyethylene glycol exposure, an osmotic control. However, in severe injury where hypoxic injury was prolonged to 15 h, P188 was only able to restore metabolic activity.

To recapitulate disruption of the blood-brain endothelium monolayer, Inyang et al. assessed permeability across murine brain endothelial monolayers that were subjected to blast-induced microcavitation MMP-2 & 9 expressions were significantly upregulated in endothelial cells that were exposed to TNF-alpha or microcavitation [17]. Furthermore, an inverse relationship between MMP-2 & 9 and tight junction protein (ZO-1) expression was noted, suggesting that compromised blood-brain barrier after bTBI may be due to elevated levels of MMP-2 & 9 leading to decreased ZO-1. Interestingly, P188 was able to inhibit MMP-2 & 9 expressions while upregulating ZO-1, thus attenuating blood-brain barrier permeability and restoring tight junctions. Although the relationship between P188 and MMP-2 & 9 is still unclear, the authors point to studies that suggest P188 may inhibit NF-kB signaling, which plays a key role in MMP transcription [57].

### 2.3. Neuronal Cells and Ischemia Reperfusion-Based Models

Meyer et al. investigated the effects of post-conditioning with P188 on mouse primary isolated cortex neurons in a combination of 5-hr hypoxia and compression followed by 2-hr reoxygenation model. Serious injury was titrated to see significantly reduced cell viability and metabolic activity; however, no significant response to P188 treatment was noted [18]. This finding is in contrast to many other in vitro studies of neurons being subjected to IR and mechanical trauma which showed P188-mediated neuroprotection through the repair of neuronal plasma membranes, protection against excitotoxicity and oxidative stress, and a decrease in mitochondrial permeabilization [19,20,21,22,26,27,58,59].

Luo et al. studied the effect of P188 on primary cortical neurons (PCNs) exposed to a combination of simulated ischemic injury and glucose deprivation [19]. PCNs that were treated with P188 had decreased mitochondrial cytochrome c release leading to a decrease in caspase-3 activation. P188 also reduced the expression of autophagy markers LC3-II and Beclin-1, both of which were markedly increased in injured PCNs. The authors of this study were able to demonstrate that P188’s neuroprotective effects in PCNs can be partially explained through inhibition of the autophagic pathway and cytochrome c release into the cytosol.

Gu et al. investigated P188 effects in an in vitro model of simulated cerebral IR and glucose deprivation injury in mouse cultured hippocampal HT22 neurons in addition to an in vivo mouse middle cerebral artery occlusion model [20]. Using Triton X-100, a membrane permeabilizing detergent, the authors showed P188 membrane resealing properties in vitro. The authors also demonstrated that brain tissue from in vivo injury contained elevated MMP-9, which was attenuated by P188 treatment. While this final fact was not shown in vitro, other studies have implicated MMP-9 in degeneration of the blood-brain barrier and which is reversed by P188 through its inhibition of the NF-kB pathway [17], p188. However, further research is indicated to characterize the exact mechanism by which P188 leads to MMP-9 inhibition.

### 2.4. Neuronal Cells and Mechanical Injury Based Models

Serbest et al. were able to demonstrate that P188 administration led to increased cell survival of PC2-derived neuronal cells in a controlled cell shearing device model of mechanical injury [21]. In a follow-up study, they showed that P188 neuroprotection is achieved through inhibition of p38 MAPK which prevented neuronal apoptosis and necrosis [58]. Interestingly, administration of a p38 selective inhibitor was much less effective at rescuing neuronal cells, suggesting that P188 may be acting upstream of p38 and/or involves additional signaling pathways.

Luo et al. evaluated the role of P188 in mitochondrial and lysosomal membrane permeability using an in vitro cell shearing device (VCSD) model of mechanical trauma in cultured primary neurons [22]. P188 mitigated loss of the mitochondrial membrane potential and inhibited cytochrome c release from mitochondria. P188 also decreased cathepsin B release from lysosomes which is implicated in initiating cell death through the conversion of p22 Bid to p15 tBid; however, the details of this apoptotic signaling pathway in VCSD are still unclear.

Kilinc et al. showed that fluid shear stress in embryonic chick forebrain neurons led to diffuse axonal injury characterized by axonal beading [59]. They attributed this diffuse axonal injury (DAI) to increased intracellular calcium levels leading to increased calpain activity, a calcium-dependent cysteine protease, which causes apoptosis through degradation of the neuronal cytoskeleton and membrane proteins. The pattern of axonal beading corresponded to the location of mitochondria along the length of the axon, implicating impaired mitochondrial function in DAI pathology. Interestingly, both intracellular calcium and calpain activity was inhibited by P188; however, further studies are required to understand how P188 interacts with calcium and calpain as well as mitochondria in order to reverse mechanical trauma-induced DAI [23].

Yildirim et al. used propidium iodide, a membrane impermeable fluorescent dye, to study the effect of P188 on cortical spreading depression, a pathological process seen in TBI that is characterized by megachannel opening leading to the propagation of altered brain activity [24]. P188 was shown to inhibit this megachannel opening in the mouse brain cortex and hippocampal dentate gyrus neurons, likely due to its membrane sealing properties.

Pille and Riess were unable to find any effect of P188 on rat forebrain mitochondria that were subjected to oxidative stress through in vivo asphyxia cardiac arrest or in vitro hydrogen peroxide exposure [25]. In contrast, the only other known study which investigated the effect of P188 on isolated mitochondria found that P188 inhibited mitochondrial outer membrane permeabilization in mitochondria that were exposed to tBid [60]. Further studies investigating the effect of P188 on isolated mitochondria subjected to different stressors are indicated.

Marks et al. investigated the effects of P188 administration on N-methyl-D-aspartate-induced excitotoxicity and oxidative stress in hippocampal and cerebellar neurons [26]. The authors were able to determine that P188 inserts itself into the plasma membrane by measuring an increase in whole-cell capacitance. P188 also blocked lipid peroxidation, suggesting that it may function as an ROS scavenger.

In a follow-up study to their in vivo investigations on TBI and P188, Bao et al. showed that P188 led to increased autophagy activation in rat PC-12 cell models of TBI [27]. The authors measured an increased Beclin-1/Bcl-2 and LC3II/LC31 ratios as well as downregulation of p62 in mice and cells that were treated with P188. Further studies are required to understand the relationship between P188 and autophagy regulation in various preclinical TBI models and to determine whether autophagy is a pathological or controlled neuroprotective process [61].

## 3. Future Directions

One possible direction for future study is to utilize the suspected mechanism of action of P188 within the cell membrane bilayer to develop a molecule with improved binding to the exposed pore, higher frequency of binding to pores of various sizes, or tighter hydrophilic grip of the exposed P188 block. New polymers are being explored to this end, converting from the ‘tri-block’, core and two arm units, to a ‘di-block’, core and one arm unit. Simultaneously, varying the length of either the core or the arm may alter the affinity of the di-block for cell membrane holes. In mouse coronary artery endothelial cells exposed to IR injury, di-block efficacy varied, with all performing at least on-par with P188. Here, the PPO size ranged from 10 to 20 units, as compared with 30 of P188 and PEO arm length from 113 to 226 units, longer than P188’s 75 [62].

Additionally, expansion of 3D cell culture approaches to in vitro investigation may more closely mimic the clinical state. One method that is utilized for this approach is that of co-cultures, where two cell types are brought into near physical contact with nested wells, allowing crosstalk, but isolation of cell lines for separate downstream analysis. Cell cross talk may be important in IR and TBI injury and recovery [63]. In an endothelial cell and cardiomyocyte co-culture with IR injury, endothelial cell quantity affected cardiomyocyte protection both with and without the addition of P188 [64]. In a more relevant model for TBI, astrocyte-endothelial cell co-culture, mimicking a blood-brain barrier, may affect injury as a potential positive feedback loop between endothelial cells and astrocytes in response to inflammation may exist [54]. Alternatively, in recovery from injury, astrocytes perform cross-talk with nearly all cell types and are a major driver for neurogenesis and angiogenesis [65]. In opposition to cardiac insult, co-culture of a blood-brain barrier model may increase injury and further heighten P188’s protective effects.

While continued investigation of the mechanism behind P188 necessitates in vitro models, simultaneous expansion of in vivo modeling of TBI is pertinent. Treatment protocols must be assessed in vivo but first a reliable and properly classified injury in an animal model is required. Continued model discrepancies will only lead to further muddled data on the utility of therapeutics in TBI.

## 4. Conclusions

P188 is a long-studied molecule for membrane stabilization with a high level of safety in clinical studies. Its proposed utility ranges throughout cardiac, neurologic, and even musculoskeletal tissues and against a variety of insults from IR to inherited defects. Membrane integrity is essential to cellular function, and damage to the membrane is a down-stream effect of many injurious processes. However, the full mechanism of P188’s action has yet to be fully elucidated, and the in vitro studies to date on TBI rely on mixed endpoints and demonstrate varied efficacy. Furthermore, TBI is a disease of varied etiology and outcomes with a combination of insults from ischemia reperfusion to axonal transection, to compressive injuries. Through increasing investigation complexity, it has been shown that both grey and white matter may be affected in TBI with distinct pathophysiologic mechanisms which may require separate or multidrug therapies to adequately address [66]. P188 may help stabilize cellular function, buying time for intrinsic repair mechanisms to catch up in severely injured cells. Given the high burden TBI and the lack of efficacious treatments to date, expeditious continued evaluation in advanced pre-clinical and even clinical models is warranted to discover new therapeutic targets and drugs and to push them towards clinical utility.

## Figures and Tables

**Figure 1 ijms-24-03334-f001:**
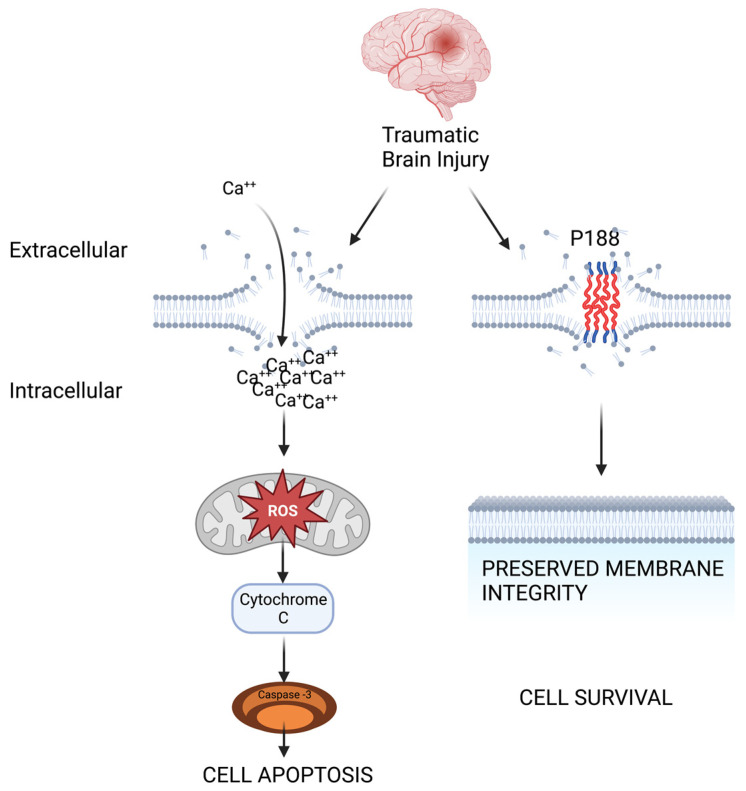
Schematic depicting how P188 mitigates TBI-induced cellular membrane permeability, thus preventing dysregulated calcium influx, reactive oxygen species (ROS) formation, and activation of apoptotic signaling cascades.

**Table 1 ijms-24-03334-t001:** Mechanism by which P188 confers protective effects in different cell types/in vitro disease models.

Author	Disease Model	Cell Type	P188 Protective Mechanism
Non-TBI models treated with P188			
Sandor et al. [8]	Vaso-occlusive crisis due to sickle cell anemia	Normoxic and hypoxic RBCs	↓Blood viscosity, ↓RBC aggregation, ↓endothelial cell adhesion
Yasuda et al. [9]	Duchenne Muscular Dystrophy	Dystrophic cardiomyocytes	↓stretch-mediated calcium overload, ↓cardiac passive tension
Salzman et al. [10]	Ischemia-reperfusion injury	Adult mouse cardiomyocytes	cell membrane repair, ↓calcium influx
Salzman et al. [11]	Ischemia-reperfusion injury	Rat cardiomyocytes	↑nitric oxide synthase, ↓infarct size
Eskaf et al. [12]	Ischemia-reperfusion injury	Rat cardiomyocytes	↔mitochondrial function (ATP synthesis)
Bartos et al. [13]	ST-elevation myocardial infarction	Pig cardiomyocytes	↑mitochondrial viability, ↓oxidation, ↓infarct size, ↓troponin leak
TBI models treated with P188: Astrocytes			
Kanagaraj et al. [14]	Blast-induced TBI	Mouse C8-D1A Astrocytes	↑cell viability, ↑calcium handling, ↓ROS production
Chen et al. [15]	Blast-induced TBI	Mouse C8-D1A Astrocytes	reseal N-type calcium channels, preserve calcium spiking
TBI models treated with P188: Endothelial Cells			
Lotze et al. [16]	Ischemia-reperfusion & compression TBI	Mouse brain microvascular endothelial cells	↑cell viability, ↑metabolic activity, ↑nitric oxide, ↓membrane damage
Inyang et al. [17]	Blast-induced TBI	Mouse brain microvascular endothelial cells	↓MMP-2 & 9, ↑ZO-1, restore tight junctions
TBI models treated with P188: Neurons (Ischemia-reperfusion models)			
Meyer et al. [18]	Ischemia-reperfusion & compression TBI	Mouse primary cortical neurons	↔cell viability, mitochondrial viability, membrane damage, caspase-3 activity
Luo et al. [19]	Ischemic injury and glucose deprivation	Mouse primary cortical neurons	↓mitochondrial cytochrome c, ↓caspase-3, ↓LC3-II, ↓Beclin-1
Gu et al. [20]	Ischemic injury and glucose deprivation Middle cerebral artery occlusion	Mouse hippocampal HT22 neurons Mouse whole brain	↑membrane resealing ↓MMP-9
TBI models treated with P188: Neurons (Mechanical Injury based models)			
Serbest et al. [21]	Cell-shearing device model	PC2 derived neuronal cells	↑cell survival, ↓p38 MAPK
Luo et al. [22]	Cell-shearing device model	Cultured primary neurons	↓mitochondrial cytochrome c release, ↓lysosomal cathepsin B release
Kilinc et al. [23]	Fluid-shear stress model	Embryonic chick forebrain neurons	↓intracellular calcium, ↓calpain activity, ↓apoptosis
Yildirim et al. [24]	Cortical Spreading Depression	Mouse brain cortex and hippocampal dentate gyrus neurons	↓megachannel opening
Pille et al. [25]	In vivo asphyxia cardiac arrest and in vitro hydrogen peroxide exposure	Rat forebrain neurons	↔mitochondrial viability
Marks et al. [26]	Excitotoxic and oxidative injury	Rat hippocampal and cerebellar neurons	↓lipid peroxidation, ↓intracellular content loss
Bao et al. [27]	Scratch TBI Left-hemispheric drop-weight TBI	Rat PC-12 cells Mouse CD1 cortex and hippocampus	↑wound healing rate ↑Beclin-1/Bcl-2, ↑LC3II/LC31 ratios, ↓p62

↑increase, ↓decrease, ↔no change.

## Data Availability

Not applicable.

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
