# Peer review of "P188 Therapy in In Vitro Models of Traumatic Brain Injury"

_ijms, 2023, doi:10.3390/ijms24043334_

Round 1

Reviewer 1 Report

Thank you for submitting your paper on P188 and in vitro models of TBI. Below are my comments:

Abstract

- The second sentence spanning lines 13-15 is not necessary as this is a paper discussing in vitro models of TBI

Introduction

- There is a sufficient overview of TBI pathophysiology. However, lines 64 through 80 introduce treatment options for TBI in an abrupt way. I recommend this section be moved after an introduction to in vitro models and before P188 discussion.

- Prior to section 2.1, it would be helpful to lay out the organization of the paper. You include a sentence to that effect on lines 108-109. I would build on that at the end of the introduction to improve organization.

Sections 2.1 and 2.2

- The organization is difficult to follow here. Introducing P188 prior to in vitro models of TBI does not flow well. I recommend introducing in vitro models of TBI then discussing P188. In essence, swap sections 2.1 and 2.2

- Lines 95-102 do not have relevance to a neuroscience model. I appreciate wanting to provide good background on P188 but this distracts the reader from the main topic of the paper.

- Lines 116-118 you mention the increased precision of in vitro TBI models without discussing a loss of accuracy and difficulty with translation into the clinic. I would add a comment on this and briefly discuss the variables that the cited authors see as "confounding" to clarify your point.

2.3 Outcome Measures

- This section is very short and can be included with the preceding section In Vitro Models of Traumatic Brain Injury

Section 3

- Section 3.1, lines 156-165 do not mention astrocytes once despite the title of the section. The introduction of blast-induced TBI without any discussion of astrocytes under this section does not flow well. I would consider removing these lines and including more detail about astrocytes.

- Section 3.2 is good

- Section 3.3 lines 222-223 mention future studies are needed to determine the role of autophagy as pathological or controlled neuroprotective process. I would qualify it by adding appropriate references as the literature includes references showing autophagy reduces post-traumatic complications and chronic inflammation as well as showing some impairment when acutely stimulated in certain contexts.

Section 4 - Future Directions

- Lines 292-294 are not relevant and distract the reader

- This section is too short. It briefly discusses P188 and next directions but can include more author speculations and suggestions for the field with respect to in vitro models for TBI. This is vaguely mentioned with 3D and co-cultures but this can be elaborated to improve the paper. For example, how does the inclusion of vasculature or another cell type affect P188 efficacy?

Conclusion

- I would be more circumspect about P188's chances given the many failures cited earlier in the paper and mixed evidence presented throughout.

Reviewer 2 Report

This review article provides a comprehensive overview of the current literature on in vitro models of TBI treated with the copolymer and surfactant Poloxamer 188 (P188). The article presents a balanced view of recent work in this subject area and the topic should be of interest to the audience of this journal and beyond, as in my opinion, P188 has still to reveal more of its yet unrecognized and unknown properties and mechanistics that can broadly potentially benefit all forms of cellular injury. Despite a large body of research using predominantly in vivo and in vitro models of brain injury, there is no medication that can reduce brain damage or promote brain repair mainly due to our lack of understanding in the mechanisms and pathophysiology of the TBI. 

From what I personally researched, the bibliography is up to date and sufficient as to the use of P188 in in vitro models and in my opinion the manuscript provides a very balanced view of the work by active groups in the subject area.  Such a paper can make a major impact on the research field covered, as it nicely condenses all the important in vitro work in this field and critically applies comparison, recommendations, analysis and opinion to those studies when protection afforded by P188 is variable and comes with mixed endpoints and results.

Such reviews can make an impact through enabling the development of newer cellular models and technologies by providing to help other researchers perform similar work.

The paper is easy to follow, clear and free from grammatical or spelling errors. The abstract, future directions and conclusions are adequate and to the point, but here I would like to mention some points that I would like the authors to include inorder to further strengthen their paper:

1.      In the concluding section, the authors might want to expand on the theme that TBI is a disorder with different etiologies, variable severities of injury and outcome with a combination of injury to both grey and white matter elements.  Moreover, it is by now well established that both regions are effected by distinct pathophysiological mechanisms that might be required to be treated separately or using combinational therapy as the ‘dream’ of using a single ‘wonder’ drug is far from real.

2.      P188 as a multiblock copolymer surfactant, has been shown to protect against ischemic tissue injury of cardiac muscle, testes and skeletal muscle, but the mechanisms have not been fully understood and likewise the neuroprotection afforded by P188 in TBI comes with mixed endpoints and results. For this reason, TBI has to considered as being an ongoing disease process and thus the much wanted urgency for a high-throughput system to be able to discover more effective and precise therapeutic targets for TBI.

3.      In vitro studies are important to dissect out mechanistics, but ultimately in vivo models are much preferred as the true and real outcome of any insult from any treatment regime can only be reliably assessed through functional imaging, gain of function and behavioural studies in terms of outcome and scores. On the same lines, advancement to develop more reliable in vivo animal models of TBI is urgently needed as it is a well known fact that in vivo models of TBI come with significant discrepancies from study to study and which are not properly classified in terms of the degree of injury (mild,moderate, severe).

4.      This manuscript would gain more in terms of exposure and be very helpful to researchers conducting similar studies in vitro, wherein a Table is inserted within the text that summarizes the key published experimental findings cited in this paper, including endpoints, model systems etc that made use of P188.

5.      The manuscript would benefit to the readership of this journal wherein a schematic or cartoon were to be presented to show the putative neuroprotective strategies/mechanisms that are involved in the use of P188 in TBI in addition to the well-known membrane resealing potential.

Minor

Line 18: to include excitotoxicity

Lines 49, 302: to substitute ‘cutting’ with transection or axotomy

Line 103-105, requires references

Round 2

Reviewer 1 Report

Thank you for addressing my comments from the previous round of revisions. Well done!